# Properties of CrO$_x$/MCM-41 and Its Catalytic Activity in the Reaction of Propane Dehydrogenation in the Presence of CO$_2$

Maria Igonina [1], Marina Tedeeva [1], Konstantin Kalmykov [1], Gennadiy Kapustin [2], Vera Nissenbaum [2], Igor Mishin [2], Petr Pribytkov [1,2], Sergey Dunaev [1], Leonid Kustov [1,2,*] and Alexander Kustov [1,2,*]

[1] Chemistry Department of M. V. Lomonosov, Moscow State University, 1–3 Leninskiye Gory, Moscow 119991, Russia
[2] N. D. Zelinsky Institute of Organic Chemistry, Russian Academy of Sciences, 47 Leninsky Prospect, Moscow 119991, Russia
[*] Correspondence: lmk@ioc.ac.ru or lmkustov@mail.ru (L.K.); kyst@list.ru (A.K.)

**Abstract:** Propylene is an important raw material for the production of many valuable compounds, especially polypropylene, the consumption of which continues to grow every year. The reaction of oxidative dehydrogenation of propane, where carbon dioxide is used as a mild oxidant, is a promising method for producing propylene. At the same time, the problem of utilization of greenhouse gas CO$_2$ is partially solved. The synthesis and analysis of the physicochemical properties of mesoporous silicate MCM-41 and supported catalysts CrO$_x$/MCM-41 prepared on its basis were carried out. These catalysts were prepared using incipient wetness impregnation. The support and catalysts were characterized by the methods of low-temperature nitrogen adsorption, TG-DTA, XRD, SEM, TPR-H$_2$, UV/Vis diffuse reflectance spectroscopy, and small-angle X-ray scattering. It is shown that chromium is present in the samples simultaneously in the form of Cr$^{3+}$ and Cr$^{6+}$. The catalytic tests were performed in the range of 550–700 °C. The highest selectivity for propylene was observed for the 5%Cr/MCM-41 catalyst and was 76% at a temperature of 650 °C with a propane conversion of 20%. The deposited catalysts Cr/MCM-41 and Cr/SiO$_2$ (Acros) were compared. The propylene selectivity for the MCM-41-supported catalyst was ~1.5 times higher than that for the SiO$_2$-supported catalyst.

**Keywords:** mesoporous silicates MCM-41; chromium catalyst; oxidative dehydrogenation of propane; propylene; carbon dioxide

## 1. Introduction

Currently, the oxidative dehydrogenation of light alkanes in the presence of CO$_2$ is attracting great attention due to technological and environmental advantages [1–3]. The low yield of alkenes in comparison with conventional cracking does not allow the widespread use of oxidative dehydrogenation; however, this problem can be solved by developing and using catalysts with effective characteristics [4–8].

When carbon dioxide is used as an oxidizing agent in the alkane dehydrogenation reaction, the active sites of the catalyst are constantly regenerated [9,10]. During the catalytic dehydrogenation reaction in the absence of an oxidizing agent, oxygen in the catalyst lattice participates in the oxidation of H$_2$ with the formation of water [11–14]. The rate of water formation decreases with a gradual increase in the yield of molecular H$_2$, which shifts the thermodynamic equilibrium toward the reagent, contributing to a decrease in the light alkane conversion, whereas CO$_2$ induces the reverse water gas shift reaction (CO$_2$ + H$_2$ → CO + H$_2$O), thereby facilitating the removal of thermodynamic restrictions [15–17].

One of the promising reactions of oxidative dehydrogenation of light alkanes is the reaction of oxidative dehydrogenation of propane (ODP) in the presence of CO$_2$ to produce propylene. Propylene is an important starting chemical raw material in the production

of valuable monomers and polymers, such as acetone, cumene, propylene oxide, acrylonitrile, acrylic acid, acrolein, and polyacrylonitrile. Propylene is particularly important for the production of polypropylene, which is produced on a large scale, more than 50 million tons per year.

ODP in the presence of $CO_2$ was studied on various catalytic systems containing mainly transition metals V, Cr, Fe, Co, Ni, Zn, and Mo, as well as Ga, In, and Au [18–23]. The most active in ODP are catalysts based on chromium oxide [16,24–29]. To obtain a more active catalyst based on chromium oxide, a high dispersion of the chromium oxide particles must be achieved. For this reason, more attention is paid to highly porous supports, including mesoporous materials such as SBA-1, SBA-15, MCM-41, MSU-x, and MSS-x [30–35]. One of the promising mesoporous materials is MCM-41—mesoporous molecular sieves that have uniform and well-ordered mesoporous channels with controlled pore sizes from 2 to 10 nm [36–38], as well as a large surface area (about 1000 m$^2$/g) [39]. In view of this, they can be used as promising supports for Cr-containing catalysts. When conducting ODP using $CO_2$ on Cr/MCM-41 catalysts, high selectivity values for propylene up to 88% were achieved with propane conversions of 15–30%. However, such high selectivity values are achieved with significant dilution of the reaction mixture with inert gases such as helium and nitrogen or with significant dilution of the reaction mixture with an excess of carbon dioxide (a mixture of $C_3H_8$:$CO_2$ at a molar ratio from 1:3 to 1:7), while the catalyst loading is usually 0.4–0.5 g [40,41]. In addition to the conversion of feedstock and selectivity for target products, it is also possible to calculate the productivity of the catalyst. The productivity of the catalyst is an important characteristic when using catalysts in industry, because it allows one to evaluate the efficiency of the unit mass of the catalyst relative to the unit mass of the feedstock. The aim of this work was to study the stability of the Cr/MCM-41 catalysts, as well as regeneration, the effect of the chromium content on the structure, morphology, and catalytic activity of these catalysts, and the effect of reaction conditions on the productivity of the catalysts. For this purpose, mesoporous molecular sieves MCM-41 were synthesized, and catalytic systems with contents of 1, 3, 5, 7, and 9 wt.% Cr were obtained on their basis. These catalytic systems were investigated in the reaction of propane dehydrogenation with the participation of $CO_2$ as a mild oxidant under conditions closest to industrial ones. For this purpose, larger catalyst weights were taken and a minimal excess of carbon dioxide relative to propane was used during the reaction.

## 2. Results

### 2.1. BJH/BET Measurements

The synthesized mesoporous silicate MCM-41 and the Cr/MCM-41 catalysts were studied using low-temperature nitrogen adsorption. Adsorption isotherms are shown in Figure 1a. All of the silica materials exhibited type IV isotherms according to IUPAC classification. The MCM-41 isotherm showed an H4 type hysteresis loop, indicating the presence of micropores in the material under study. The absence of a desorption hysteresis loop during nitrogen adsorption is typical for MCM-41 samples with a pore size of 2–4 nm [42]. The pore volume distributions by size are shown in Figure 1b. The introduction of chromium oxide promoted a decrease in the pore volume and, consequently, a decrease in the surface area of the Cr/MCM-41 catalysts. At the same time, the average pore diameter decreased from 2.3 to 2.0 nm. This can indicate the deposition of a part of the chromium oxide particles in the pores of the MCM-41 support. When chromium oxide particles are applied to the MCM-41 support, the bimodal nature of the pore size distribution completely disappears. This means that the pores of the MCM-41 support in the range from 3.3 to 4.1 nm are completely filled with chromium oxide particles.

Table 1 shows the texture characteristics for the MCM-41 sample and the Cr/MCM-41 catalysts calculated from adsorption isotherms.

After supporting Cr onto the MCM-41 support, the specific surface area decreased slightly. After supporting 1% Cr onto the MCM-41 support, the pore volume and the pore

diameter decreased. With a further increase in the chromium content to 9%, these values almost did not change.

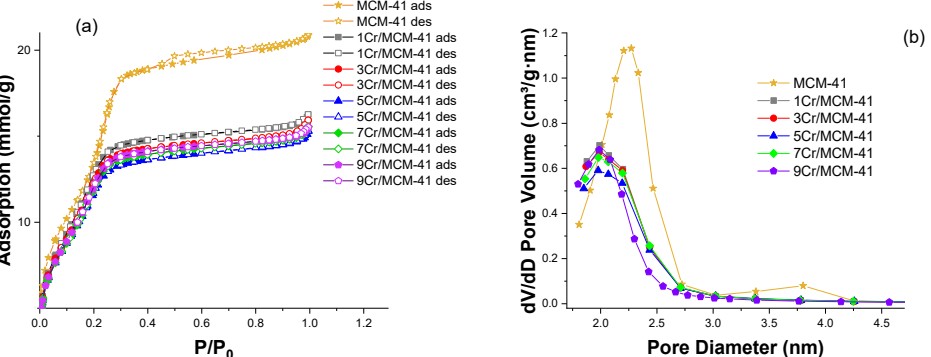

**Figure 1.** Nitrogen adsorption/desorption isotherms (**a**) and the pore volume distributions by size (**b**) for the MCM-41 support and the Cr/MCM-41 catalysts.

**Table 1.** Textural characteristics of the MCM-41 support and 9 Cr/MCM-41 catalyst.

| Sample | $S_{BET}$, $m^2/g$ | $V_{meso}$, $cm^3/g$ | $D_{pore}$, Å |
|---|---|---|---|
| MCM-41 | 1260 | 0.7 | 23 |
| 1 Cr/MCM-41 | 1137 | 0.56 | 20 |
| 3 Cr/MCM-41 | 1092 | 0.55 | 20 |
| 5 Cr/MCM-41 | 992 | 0.53 | 20 |
| 7 Cr/MCM-41 | 1018 | 0.54 | 20 |
| 9 Cr/MCM-41 | 969 | 0.52 | 20 |

### 2.2. Thermogravimetry and Differential Thermal Analysis (TG-DTA)

The TG-DTA method was used to select the temperature range of heating samples that ensured effective formation of the catalyst. First of all, the process of removing the template from synthesized MCM-41 dried at a temperature of 100 °C for 24 h was investigated (Figure 2a).

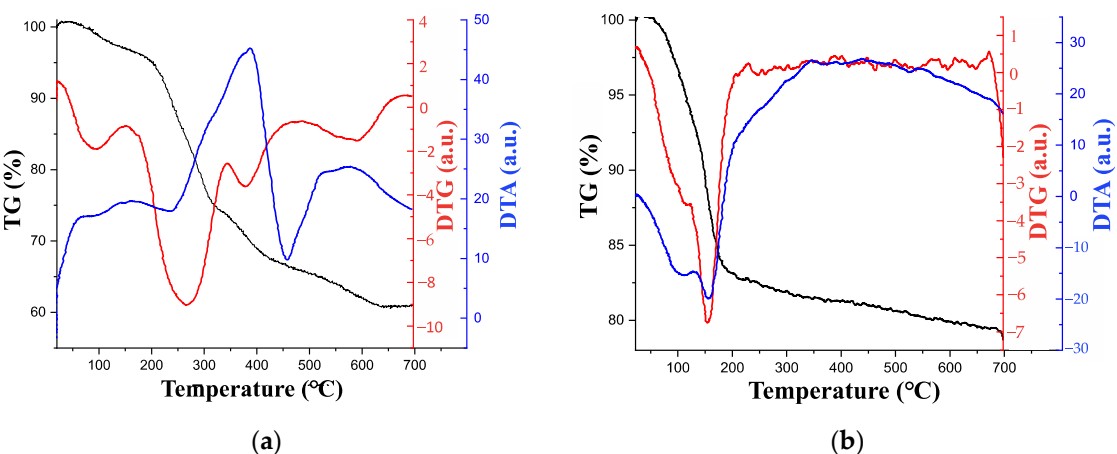

**Figure 2.** Derivatograms of (**a**) the MCM-41 carrier with a template and (**b**) the 9 Cr/MCM-41 catalyst dried in air at a temperature of 100 °C for 24 h.

The process of removing the template from a freshly prepared sample MCM-41 (obtained after drying at a temperature of 100 °C for 24 h) was investigated using the TG-DTA method (Figure 2a).

The observed low-temperature weight loss, up to 150–200 °C, was due to the removal of alcohol used for washing and desorption of water. The next stage of the weight change was associated with the desorption of the template from the pores (a minimum on the DTG curve at 265 °C) and its oxidative degradation in the temperature range of 300–490 °C, accompanied by an exothermic effect with a maximum at 380 °C. At temperatures above 550 °C, no noticeable change in weight was recorded, which indicates the complete removal of the template, consistent with the literature data [36]. Thus, for calcination of the carrier in air, taking into account a certain temperature margin, it is necessary to use a temperature of 550–600 ° C. Furthermore, the process of decomposition of chromium nitrate on the surface of the MCM-41 support (Figure 2b) was investigated. According to the DTG curve, the temperature of 180 °C corresponds to the removal of most nitrate ions, and complete removal of chromium nitrate occurs at 300 °C.

### 2.3. Scanning Electron Microscopy (SEM) with Energy-Dispersive X-ray Spectroscopy (EDS)

The structure of the synthesized mesoporous silicate and Cr/MCM-41 catalysts was studied by scanning electron microscopy (Figure 3). The first image shows a surface of the support, which could be used to judge the spherical shape of MCM-41 particles. The sizes of the formed crystallites were in the range from 0.2 to 1 μm and were, on average, about 0.8 μm. Cr/MCM-41 catalyst images show that the MCM-41 support particles retained their spherical shape and particle sizes. With an increase in the chromium content in the catalysts, the appearance of larger non-spherical chromium particles located at the boundaries of MCM-41 particles was observed. This was observed on samples with a chromium content of 5–9%, especially noticeable on 7/MCM-41 and 9/MCM-41 catalysts.

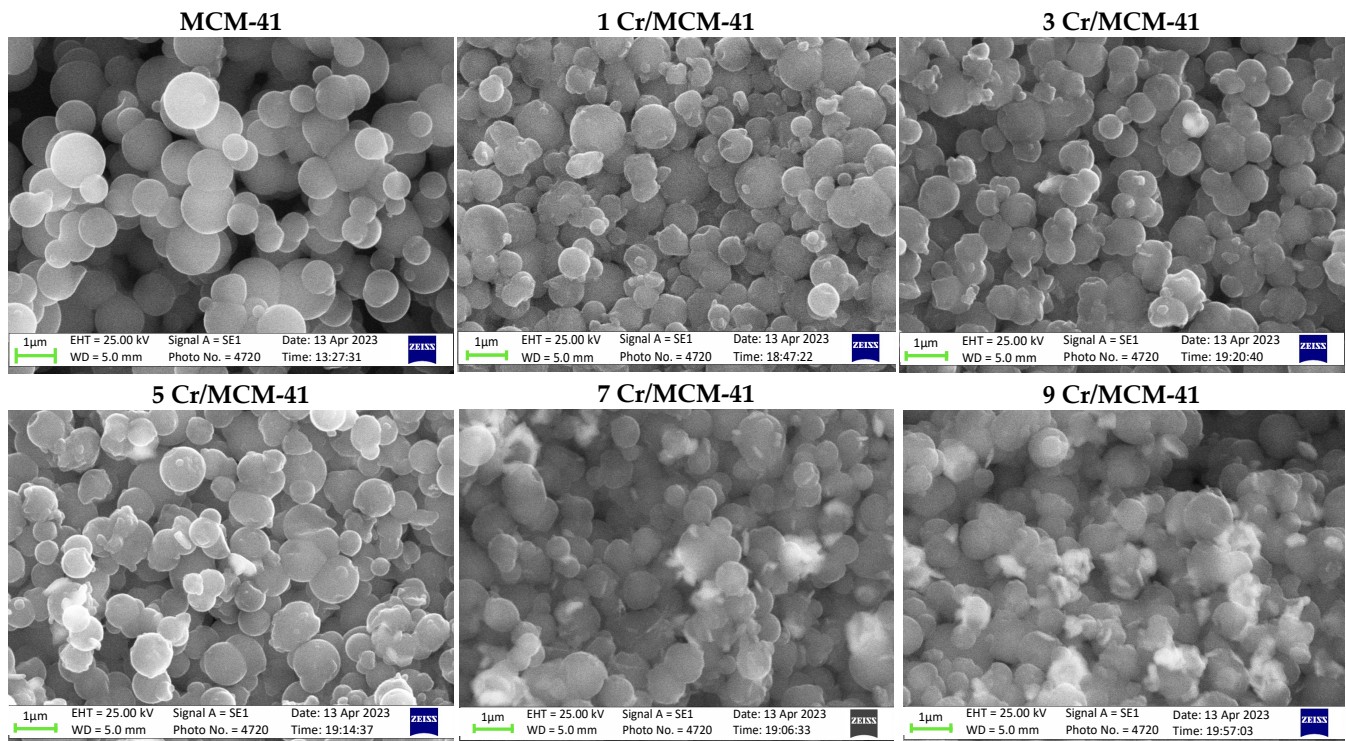

**Figure 3.** Images of the surfaces of the MCM-41 support and the Cr/MCM-41 catalysts.

Additional SEM images and the results of EDS can be found in the Supplementary Materials, Figures S1 and S2.

From the chromium mapping of the Cr/MCM-41 catalysts (Figure S1, the second row), one can judge the uniform distribution of chromium over the sample; for comparison, images of the same particles of these catalysts are shown (Figure S1, upper row).

Figure S2 shows the typical spectrum for the Cr/MCM-41 sample. When studying the surfaces of the Cr/MCM-41 catalyst samples using EDS, the ratio of elements on the samples surfaces was obtained (Table 2).

**Table 2.** The ratio of elements on the catalyst surfaces.

| Total Spectrum | Cr | Si | O | Total | Cr EDS/Cr Computational, % |
|---|---|---|---|---|---|
| 1 Cr/MCM-41 | 0.7 | 46.4 | 52.9 | 100 | 70 |
| 3 Cr/MCM-41 | 2.4 | 47.8 | 49.8 | 100 | 80 |
| 5 Cr/MCM-41 | 5.2 | 45.7 | 49.1 | 100 | 104 |
| 7 Cr/MCM-41 | 6.4 | 44.1 | 49.5 | 100 | 91 |
| 9 Cr/MCM-41 | 9.2 | 42.6 | 48.2 | 100 | 102 |

The weight percentage of Cr in 5–7 Cr/MCM-41 catalysts was in good agreement with the initial data on the content of chromium in the catalysts. In samples 1 Cr/MCM-41 and 3 Cr/MCM-41, the chromium content on the surface was slightly less than the initial data. This can be explained by the fact that the smallest chromium particles could be located in the pores of the MCM-41 support.

### 2.4. X-ray Diffraction (XRD)

A sample of mesoporous silicate MCM-41 and the catalytic systems Cr/MCM-41 were studied by XRD in the range of angles of $2\theta = 10°–60°$. The MCM-41 support and Cr/MCM-41 catalysts were also investigated in the range of angles $2\theta = 1°–7°$. Diffraction patterns are shown in Figure 4.

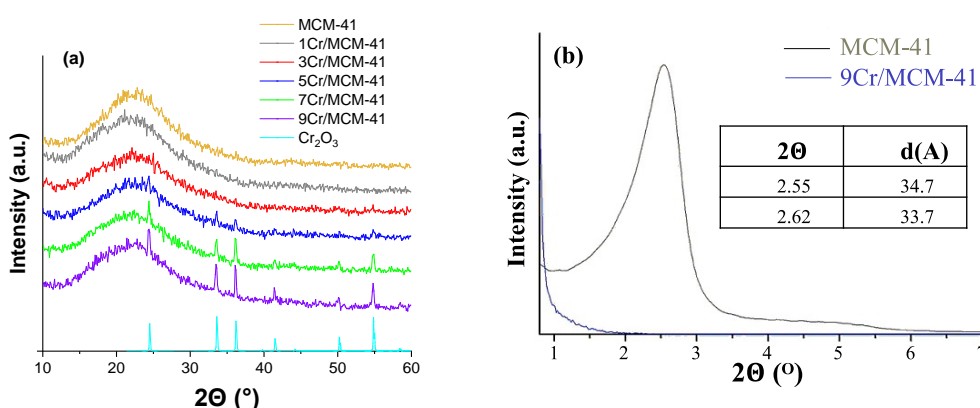

**Figure 4.** Diffraction patterns of the MCM-41 support, Cr/MCM-41 catalysts, bulk sample $Cr_2O_3$ (database) [Inorganic Crystal Structure Database (ICSD)] (**a**) and small-angle X-ray scattering data of the MCM-41 support and Cr/MCM-41 catalysts (**b**).

The sample of the MCM-41 support was X-ray amorphous (Figure 4a), there were no reflections in the range of angles $2\theta = 20°–80°$. When using the method of small-angle X-ray scattering ($2\theta < 10°$), reflections characteristic of the hexagonal structure of mesopores of the material of the MCM-41 type were clearly manifested on the diffractogram of the MCM-41 sample (Figure 4b) [37].

The diffraction pattern of the 5–9 Cr/MCM-41 catalytic systems (Figure 4a) in the range of angles $2\theta = 20°–80°$ contained diffraction lines corresponding to the crystalline phase of $\alpha$-$Cr_2O_3$. The diffraction patterns of catalytic systems with a lower chromium content lacked diffraction lines corresponding to $\alpha$-$Cr_2O_3$, which can most likely be explained by the high dispersion of chromium oxide particles on the support surface.

In the case of small-angle X-ray scattering ($2\theta < 10°$), the maximum disappeared in the diffractogram of the 9 Cr/MCM-41 catalyst. Such changes in the diffraction pattern mean that, when the carrier was impregnated with a precursor followed by thermal

decomposition, the carrier channels were partially destroyed. At the same time, a decrease in intensity in this case may also have been the result of the presence of chromium oxide particles that were heavier than the mass of silicate walls, which were randomly distributed in the pores and violated the periodicity of the lattice. The XRD data were used to calculate the interplanar distances and cell parameters for the MCM-41 sample (Figure 4b).

### 2.5. UV/Vis Diffuse-Reflectance Spectroscopy

The valence states of chromium on the surface of mesoporous silicate MCM-41 were investigated by UV/Vis diffuse-reflectance spectroscopy (Figure 5).

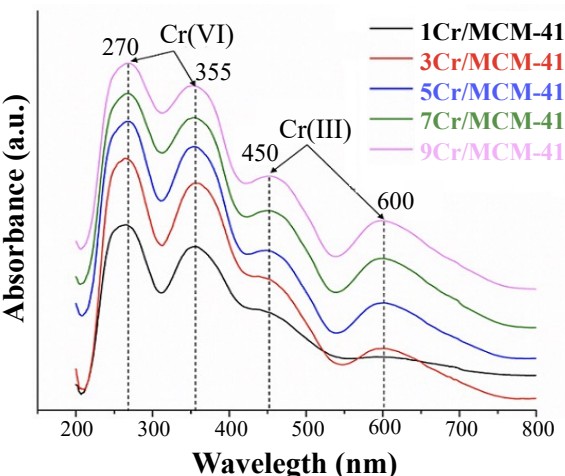

**Figure 5.** UV/Vis diffuse reflectance spectra of samples 1–9 Cr/MCM-41.

All catalytic systems exhibited four UV bands. The bands at ~270 and 355 nm corresponded to the $O_2 \rightarrow Cr^{6+}$ charge transfer transition for chromates in tetrahedral coordination. The absorption bands at 450 and 600 nm were attributed to the d–d charge transfer transition of $Cr^{3+}$ in the octahedral coordination of Cr(III) in $Cr_2O_3$ or $CrO_x$ clusters [33,43–45]. The intensity of the absorption bands increased with an increase in the concentration of chromium oxide in the samples, indicating the predominance of $Cr^{6+}$ species in catalysts with a low chromium content.

It should be noted that $Cr^{6+}$ chromium moieties on the support surface were formed from $Cr^{3+}$ species when the catalysts were calcined in an airflow. In this case, the number of $Cr^{6+}$ sites depended on the nature of the support and on the concentration of the active component; if the concentration of the active component exceeded the level of the monolayer coating, then a crystalline $\alpha$-$Cr_2O_3$ phase was formed on the catalyst surface.

### 2.6. Temperature-Programmed Reduction (TPR-$H_2$)

For the catalytic system 5 Cr/MCM-41, temperature-programmed reduction with hydrogen was carried out in the temperature range 25–850 °C (Figure 6).

The TPR-$H_2$ profile for the supported catalyst 5 Cr/MCM-41 showed a maximum in the temperature range 400–500 °C, corresponding to the reduction of $Cr^{6+}$ species that were directly related to the silica substrate. An additional low-temperature maximum was observed at 310 °C. The position of this maximum coincided well with the maximum of the low-temperature reduction observed for the calcined $\alpha$-$Cr_2O_3$, which corresponded to the reduction of $Cr^{6+}$ to $Cr^{3+}$ or $Cr^{2+}$ in $\alpha$-$Cr_2O_3$. Consequently, the presence of a low-temperature maximum confirmed the existence of $\alpha$-$Cr_2O_3$ [32,37].

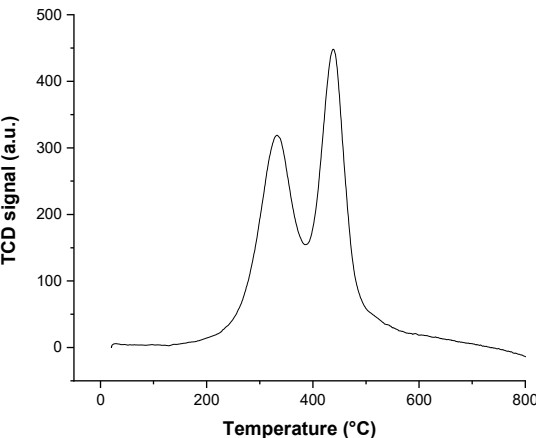

**Figure 6.** TPR-$H_2$ of the 5 Cr/MCM-41 catalyst.

### 2.7. Catalytic Activity

The synthesized catalytic systems were investigated in the reaction of catalytic propane dehydrogenation in the presence of carbon dioxide:

$$C_3H_8 + CO_2 = C_3H_6 + CO + H_2O. \tag{1}$$

In addition to the main reaction product, propylene, the formation of byproducts such as methane, ethane, and ethylene was observed.

Figure 7 shows the dependence of the catalytic activity on the content of chromium oxide in the samples.

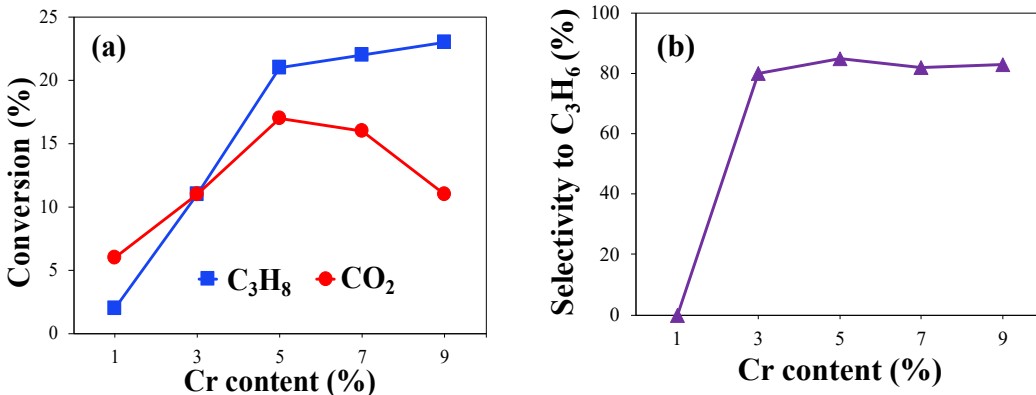

**Figure 7.** (**a**) $C_3H_8$ and $CO_2$ conversion and (**b**) selectivity for propylene for the 1–9 Cr/MCM-41 catalyst at T = 650 °C. $C_3H_8$:$CO_2$ = 1:2, he total flow rate = 30 cm$^3$·min$^{-1}$, $m_{cat}$ = 1 g.

For catalytic systems Cr/MCM-41 with a Cr content of 1–9 wt.%, at temperatures of 550–600 °C, a very low conversion of 1–3% was observed, except for 5 Cr/MCM-41, with the conversion at 600 °C being ~6%. At 650 °C, the conversion for the 3 Cr/MCM-41 sample increased above 10%, and the conversion in the case of the 5–9 Cr/MCM-41 samples increased above 20%. The highest propane conversion was observed at a temperature of 700 °C, while the maximum propane conversion (40.4%) was achieved on a catalyst with a chromium oxide content of 7 wt.%, and the lowest conversion was shown by a sample with a chromium oxide content of 1 wt.% (21%). It should be noted that the highest activity was revealed by a catalyst with a chromium oxide content of 5 wt.%; the propylene selectivity was higher than 80% at a temperature of 650 °C, while the 1 Cr/MCM-41 sample was ineffective at this temperature, and a further increase in the chromium concentration did not lead to a slight increase in the catalytic activity. With an increase in the Cr content in Cr/MCM-41 catalysts, the $CO_2$ conversion increased and reached a maximum at 5% Cr,

and then began to decrease. A decrease in the activity of catalysts with an increase in the chromium content was possible due to the formation of $CrO_x$ clusters that were not active in propane dehydrogenation in the presence of $CO_2$ [16].

Figure 8 shows the selectivities for all reaction products for the sample 1 Cr/MCM-41.

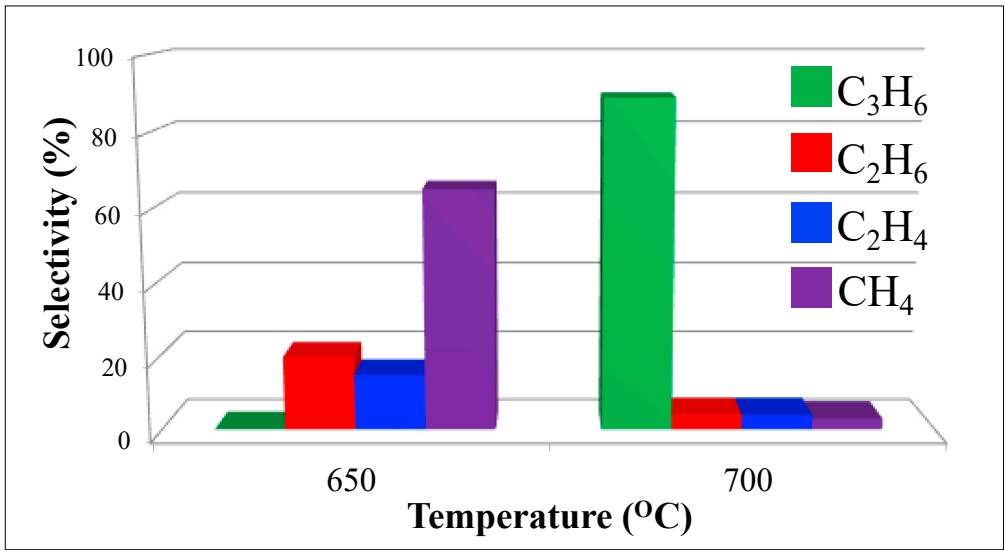

**Figure 8.** Selectivity for propylene, ethane, ethylene, and methane for the 1 Cr/MCM-41 catalyst at T = 650 and 700 °C. $C_3H_8$:$CO_2$ = 1:2, total flow rate = 30 $cm^3 \cdot min^{-1}$, $m_{cat}$ = 1 g.

The catalyst with a content of 1% Cr differed from all other Cr/MCM-41 materials. There was no formation of propylene, but substantial methane, ethane, and ethylene were formed in small quantities at a temperature of 650 °C. The formation of propylene began immediately in large quantities, the selectivity for ethane and ethylene decreased significantly, and the formation of methane occurred in trace amounts at 700 °C.

Figure 9 shows the selectivities for all reaction products for the sample 5 Cr/MCM-41.

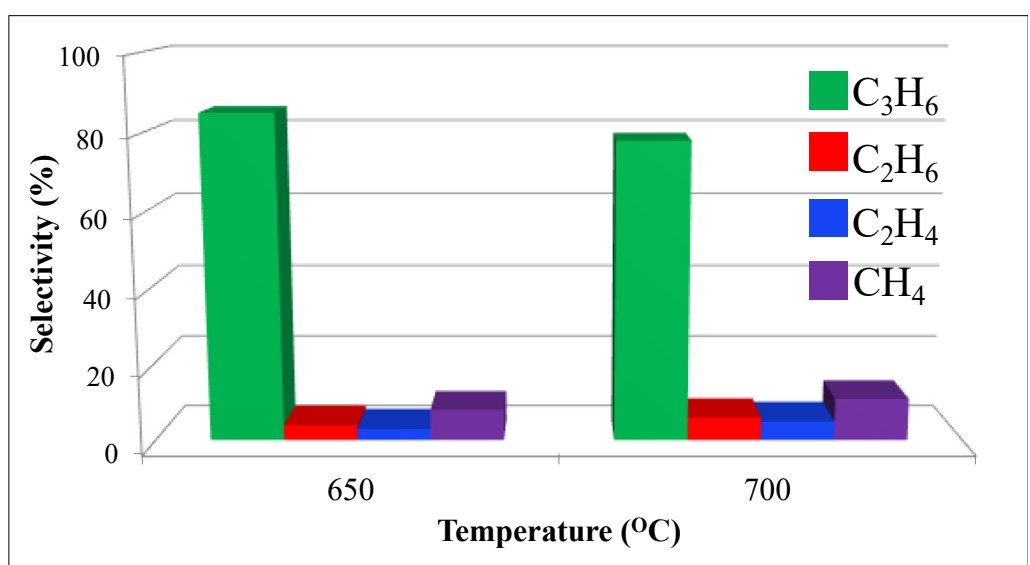

**Figure 9.** Selectivity for propylene, ethane, ethylene, and methane for the 5 Cr/MCM-41 catalyst at T = 650 and 700 °C. $C_3H_8$:$CO_2$ = 1:2, total flow rate = 30 $cm^3 \cdot min^{-1}$, $m_{cat}$ = 1 g.

The selectivity for propylene was over 80%, whereas the selectivity for ethane, ethylene, and methane was lower than 10% for the 5 Cr/MCM-41 catalyst at T = 650 °C. With an increase in temperature to 700 °C, the selectivity for propylene decreased slightly, and the

selectivity for ethane, ethylene, and methane somewhat increased. Approximately the same pattern was observed for the samples of catalysts with a content of 3, 7, and 9 wt.% Cr. The only difference was that the selectivity for methane decreased slightly for the samples containing 3, 7, and 9 wt.% Cr at 700 °C. Similar patterns were observed for the catalyst 6.8 Cr/MCM-41 in [40], which was also prepared by the incipient wetness method. With an increase in temperature, the selectivity for propylene decreased, and the selectivity for ethane and methane increased slightly. The selectivity for ethylene increased more noticeably [40].

It should be noted that, according to the results of the catalytic activity tests and the data of physicochemical analysis, in order to achieve a high efficiency of chromium oxide catalytic systems in the reaction of propane dehydrogenation in the presence of $CO_2$, it is necessary to achieve a balance of $Cr^{6+}$ and $Cr^{3+}$ species.

Previously, we investigated the Cr/SiO$_2$ (Acros) catalyst in the propane dehydrogenation reaction in the presence of $CO_2$. The reaction was also carried out with a minimal excess of carbon dioxide relative to propane 2:1 and a catalyst weight of 1 g [26]. A comparison of the activities of catalysts with a Cr content of 5% for MCM-41 and the SiO$_2$ (Acros) support with a specific surface area of 557 m$^2$/g (5 Cr/SiO$_2$) is presented in Figure 10. The propane conversion for the 5 Cr/MCM-41 catalyst was lower than the conversion for the 5 Cr/SiO$_2$ catalyst, but the propylene selectivity for the MCM-41 supported catalysts was significantly higher than for the SiO$_2$-supported catalysts.

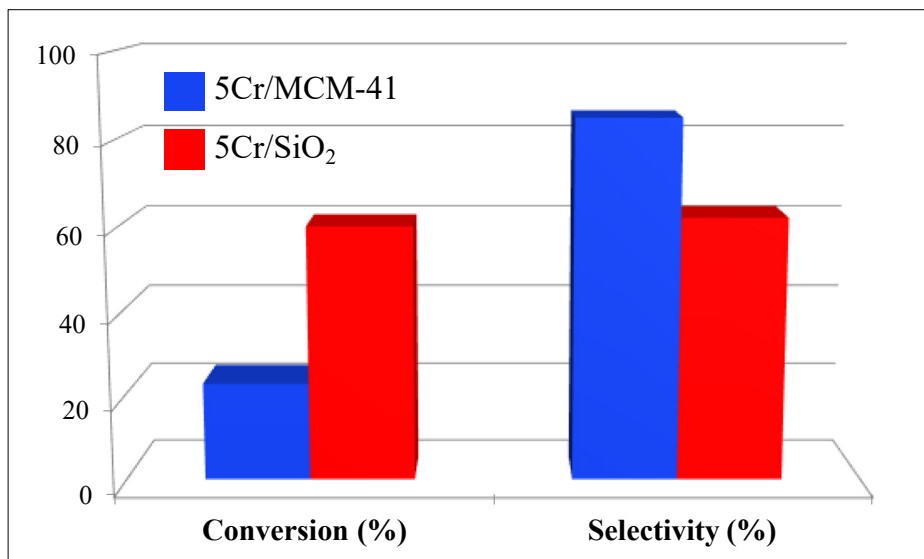

**Figure 10.** Propane conversion and propylene selectivity at 650 °C for 5 Cr/MCM-41 and 5 Cr/SiO$_2$ catalysts. $C_3H_8$:$CO_2$ = 1:2, total flow rate = 30 cm$^3$·min$^{-1}$, m$_{cat}$ = 1 g.

The stability of the 5 Cr/SiO$_2$, 5 Cr/MCM-41 and 9 Cr/MCM-41 catalysts in the reaction of propane dehydrogenation in the presence of $CO_2$ at a temperature of 650 °C was studied (Figure 11).

It should be noted that the 5 Cr/MCM-41 catalyst showed better stability. The decrease in its selectivity for propylene was 20% with continuous operation of the catalyst for 10 h, while, for the 9 Cr/MCM-41 catalyst, the decrease in propane conversion was ~45% over the same time. In the case of propylene selectivity, 5 Cr/MCM-41 also showed slightly better stability than 9 Cr/MCM-41. In 7 h, the selectivity also decreased ~ 20% and then did not change. The propane conversion on the 5 Cr/SiO$_2$ catalyst decreased by 60%, while the propylene selectivity decreased by only ~10% in the first 2–3 h, and then remained at the level of 50%. The decrease in catalyst activity was largely associated with the formation of coke on its surface, as well as with the formation of inactive chromium particles and the destruction of the mesoporous support structure [32]. To determine the cause of

deactivation, the 5 Cr/MCM-41 catalyst was regenerated after the reaction at 800 °C in an air flow for 4 h.

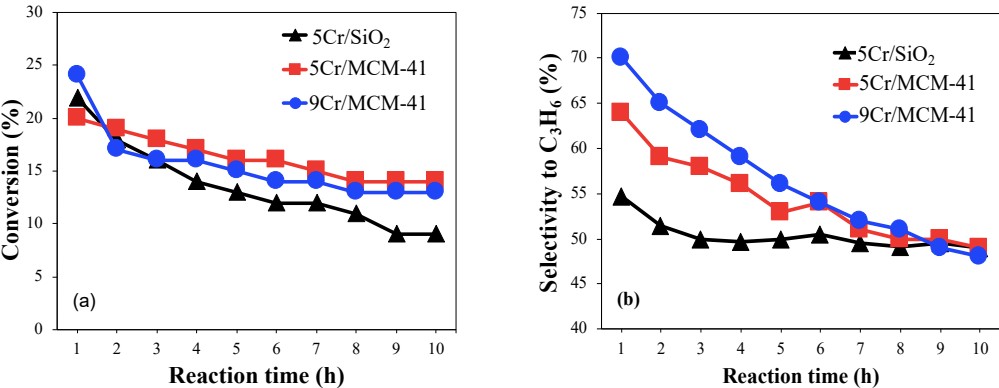

**Figure 11.** Propane conversion (**a**) and selectivity for propylene (**b**) for the 5 Cr/MCM-41 and 9 Cr/MCM-41 catalysts over time at a temperature of 650 °C. $C_3H_8:CO_2$ = 1:2, total flow rate = 30 $cm^3 \cdot min^{-1}$, $m_{cat}$ = 0.5 g.

The sample after regeneration had a slightly reduced propane conversion and selectivity for propylene, as well as a slightly reduced stability over the entire time interval (Figure 12), associated either with the formation of inactive particles or with the destruction of the mesoporous structure of the catalyst [32].

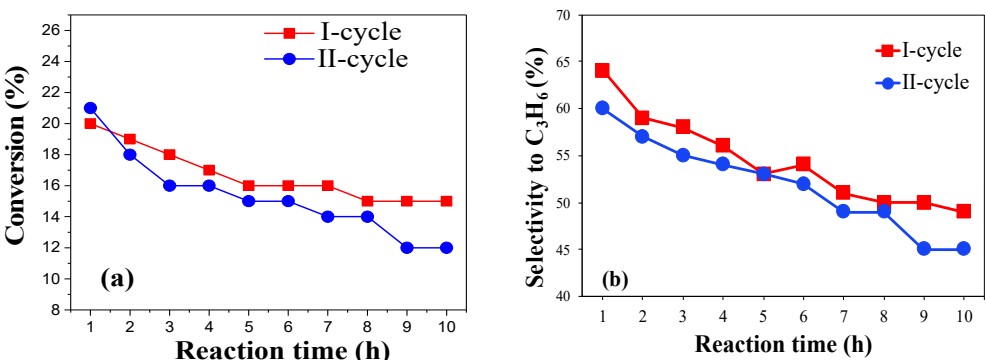

**Figure 12.** Performance after regeneration: propane conversion (**a**) and selectivity for propylene (**b**) for the 5 Cr/MCM-41 catalyst over time at a temperature of 650 °C. $C_3H_8:CO_2$ = 1:2, total flow rate = 30 $cm^3 \cdot min^{-1}$, $m_{cat}$ = 0.5 g.

Table 3 shows a comparison of the most efficient catalytic system 5 Cr/MCM-41 with literature data.

**Table 3.** Comparison of the catalytic activity of Cr-containing catalysts in the ODP-CO₂ reaction.

| Material | Experimental conditions | T, °C | Conversion of $C_3H_8$, % | Selectivity to $C_3H_6$, % | Productivity, g $(C_3H_6) \cdot kg_{cat}^{-1} \cdot h^{-1}$ | Source |
|---|---|---|---|---|---|---|
| 5 Cr/MCM-41 | $C_3H_8:CO_2$ (1:2) | 650 | 21.9 | 64 | 315 | This work * |
| 9 Cr/MCM-41 | $C_3H_8:CO_2$ (1:2) | 650 | 24 | 70 | 378 | This work * |
| Cr/MCM-41 | $C_3H_8:CO_2$ (1:3.6) | 550 | 16.7 | 90 | 286 | [46] |
| 5%Cr/SiO₂ | $C_3H_8:CO_2$ (1:3.6) | 600 | 17.7 | 100 | 202 | [47] |
| 5%Cr/SiO₂ | $C_3H_8:CO_2$ (1:3.6) | 650 | 27 | 85 | 262 | [47] |
| CrOₓ/SiO₂ | $C_3H_8:CO_2$ (1:7) | 600 | 27.7 | 90.8 | 282 | [48] |

* $C_3H_8:CO_2$ = 1:2, total flow rate = 30 $cm^3 \cdot min^{-1}$, $m_{cat}$ = 0.5 g.

## 3. Materials and Methods

### 3.1. Catalyst Preparation

The recipe for the synthesis of the MCM-41 carrier was presented elsewhere [49]. For the preparation of the MCM-41 material, the following reagents were used: 9.7 g of n-hexadecyltrimethylammonium bromide ($C_{16}$ TMABr) (Acros Organics, Geel, Belgium), 170 mL of distilled water, 68 mL of a 25% aqueous ammonia solution(Limited Liability Company "Component-Reartiv" Moscow, Russian Federation), 283 mL of ethanol (Open Joint Stock Company "Medkhimprom", Moscow, Russian Federation), and 17.3 g of tetraethylorthosilicate (TEOS) (Otto chemie pvt ltd, Mumbai, India). The molar ratio of the reagents used for the synthesis was 1 TEOS:11 $NH_3$:58 EtOH:0.32 $C_{16}$ TMABr:144 $H_2O$. Supported catalytic systems were prepared by the incipient wetness impregnation method described earlier [26]. Thus, the series of samples were obtained and marked as *X*Cr/MCM-41, where *X* = 1, 3, 5, 7, 9 wt.% of Cr (actually the samples contain the forms of $CrO_x$).

### 3.2. Catalyst Characterization

The characteristics of the porous structure of the catalysts were determined on the basis of nitrogen adsorption isotherms measured at $-296\,°C$ using an ASAP 2020 Plus unit (Micromeritics Instrument Corporation, Norcross, GA, USA). Prior to measurements, the samples were evacuated at $350\,°C$ for 3 h. The specific surface area was calculated according to the BET method, and the pore size distribution was found from the desorption branch of the isotherm via the Barrett–Joyner–Halenda (BJH) analysis. The volume of micropores was determined as the difference between the total pore volume and the cumulative volume of mesopores at 2 nm in the BJH method.

The morphology, particle size, and elemental composition of the catalyst surface were studied by scanning electron microscopy (SEM) using a LEO EVO 50 XVP electron microscope (Carl Zeiss, Oberkochen, Germany) supplied with an INCA Energy 350 energy-dispersive spectrometer (Oxford Instruments, Abingdon, Oxfordshire, England). The device allowed setting the energy of electrons in the range of 0.2–30 kV. The cathode was a heating element made of lanthanum hexaboride $LaB_6$.

Thermal analysis was carried out with a combination of thermogravimetry, differential thermogravimetry, and differential thermal analysis (TG-DTG-DTA) using a Derivatograph-C unit (MOM Szervin Kft, Budapest, Hungary). Each sample was placed into an alundum crucible and heated from 20 to $600\,°C$ in air at the rate of $10\,°C/min$. $\alpha$-$Al_2O_3$ was used as a standard; the weight of the samples was 100 mg.

X-ray diffraction (XRD) patterns of the samples were obtained with a DRON-2 diffractometer (Cu*K*$\alpha$ radiation) (Scientific and Production Enterprise "Burevestnik", Saint-Petersburg, Russian Federation). The samples were scanned in the 2θ range of 20°–70° at a rate of 1°/min. X-ray diffraction analysis at small angles was performed using an ARL X'TRA4 diffractometer (Thermo Fisher Scientific, Waltham, MA, USA) equipped with a Theta-Theta goniometer (Cu*K*$\alpha$ radiation, 40 kV, 40 mA) and an energy-dispersive detector. X-ray diffraction patterns were recorded by scanning at a speed of 0.75°/min in the range of 0.5° < 2θ < 6°. In order for the direction of the scattered beams to differ slightly from the direction of the incident beam, a special wedge-cutter of the incident beam was used. The width of the leaf slits was reduced to 0.1 mm.

Temperature-programmed hydrogen reduction (TPR-$H_2$) was performed using a semi-automatic setup supplied with a thermal conductivity detector calibrated by reduction of CuO (Aldrich-Chemie GmbH, 99+%, Steinheim, Germany) and a mixture of CuO + MgO (to be calibrated at low metal contents). The presented TPR profiles were normalized per 1 g of the material. The conditions of TPR-$H_2$ measurements were described in more detail elsewhere [50].

Diffuse-reflectance UV–Vis spectra were measured using a Shimadzu UV-3600 Plus spectrophotometer (Shimadzu Corporation, Kyoto, Japan) equipped with an ISR-603 integration sphere. The spectra were recorded at 200–800 nm at room temperature, and $BaSO_4$

was used as a standard and diluent. The weight of the catalysts was 0.1 g, and that of $BaSO_4$ was 0.5 g. The UVProbe software was used to process the spectra.

*3.3. Catalytic Activity Tests*

The dehydrogenation of propane to propylene in the presence of $CO_2$ was investigated in the temperature range 550–700 °C at an atmospheric pressure in a flow catalytic setup with a steel reactor with an inner diameter of 4 mm. The gas mixture $C_3H_8 + CO_2$ was fed into the reactor in a volumetric ratio of 1:2; the total flow rate of the gas mixture was 30 mL/min. The catalyst loading was 1 g. When studying the stability of the samples, the mass of the catalyst was 0.5 g. The analysis of the reaction products was carried out using a Chromatec-Crystal 5000 gas chromatograph equipped with a thermal conductivity detector, M ss316 3 m × 2 mm columns, Hayesep Q 80/100 mesh, and CaA molecular sieves. The temperature of the column was ramped according to the program described in [51]. The quantitative analysis was carried out using the absolute calibration method.

The $C_3H_8$ conversion and $C_3H_6$, $C_2H_6$, $C_2H_4$, and $CH_4$ selectivities were calculated using the following formulas:

$$\text{Conversion of propane (\%)} = \frac{\text{introduced propane} - \text{residual propane}}{\text{introduced propane}} \times 100, \quad (2)$$

$$\text{Selectivity of propylene (\%)} = \frac{\text{produced propylene}}{\text{introduced propane} - \text{residual propane}} \times 100, \quad (3)$$

$$\text{Selectivity of ethane (\%)} = \frac{2/3 \text{ produced ethane}}{\text{introduced propane} - \text{residual propane}} \times 100, \quad (4)$$

$$\text{Selectivity of ethylene (\%)} = \frac{2/3 \text{ produced ethylene}}{\text{introduced propane} - \text{residual propane}} \times 100, \quad (5)$$

$$\text{Selectivity of methane (\%)} = \frac{1/3 \text{ produced methane}}{\text{introduced propane} - \text{residual propane}} \times 100. \quad (6)$$

**4. Conclusions**

After supporting 1% Cr onto the MCM-41 support, the pore volume and the pore diameter decreased. With a further increase in the chromium content to 9%, these values almost did not change. The particles of synthesized MCM-41 had a spherical shape with an average diameter of ~0.8 μm. Chromium oxide particles were evenly distributed over the surface of the support. When the chromium content in the catalysts was 5% or more, the appearance of larger non-spherical chromium particles located at the boundaries of MCM-41 particles was observed. This was especially noticeable on catalysts 7/MCM-41 and 9/MCM-41. The diffraction pattern of the 5–9 Cr/MCM-41 catalysts contained diffraction lines corresponding to the crystalline phase of α-$Cr_2O_3$. It was shown that chromium was present in the states of $Cr^{3+}$ and $Cr^{6+}$ in the samples of supported Cr/MCM-41 catalysts with a Cr content of 1, 3, 5, 7, and 9 wt.%. The highest selectivity for propylene in the reaction of oxidative dehydrogenation of propane in the presence of carbon dioxide was observed for the 5 Cr/MCM-41 sample: 85% at a temperature of 650 °C with a conversion of 21%. At the same time, the propylene selectivity for the 5 Cr/MCM-41 sample was ~1.5 times higher than that for the 5 Cr/$SiO_2$ catalyst. The 5 Cr/MCM-41 showed good enough stability. The decrease in its selectivity for propylene was 20% with continuous operation of the catalyst for 10 h.

**Supplementary Materials:** The following supporting information can be downloaded at https://www.mdpi.com/article/10.3390/catal13050906/s1: Figure S1: SEM images (the upper row) and chromium (the 2nd row), silicon (the 3rd row), oxygen (the lower row) distribution maps for the Cr/MCM-41 catalysts; Figure S2: EDS of Cr/MCM-41 catalysts.

**Author Contributions:** Conceptualization, L.K. and A.K.; methodology, M.T. and P.P.; validation, K.K.; formal analysis, G.K.; investigation, M.I., I.M. and M.T.; resources, S.D.; data curation, V.N.; writing—original draft preparation, A.K.; writing—review and editing, L.K.; visualization, I.M.; supervision, A.K. and L.K.; project administration, A.K. and L.K.; funding acquisition, A.K. All authors have read and agreed to the published version of the manuscript.

**Funding:** The research related to catalyst preparation and catalytic tests was carried out with financial support from a grant of the Russian Science Foundation (project No. 20-73-10106). The study of catalysts using physicochemical methods was carried out with financial support from the Ministry of Science and Higher Education of the Russian Federation (project No. 075-15-2021-591).

**Data Availability Statement:** Data are available from the authors upon request.

**Conflicts of Interest:** The authors declare no conflict of interest.

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
