# Peer review of "Properties of CrOx/MCM-41 and Its Catalytic Activity in the Reaction of Propane Dehydrogenation in the Presence of CO2"

_catalysts, doi:10.3390/catal13050906_

Round 1
Reviewer 1 Report
In the manuscript series of CrOx/MCM-41 catalysts were prepared, characterized base physicochemical techniques and tested in the dehydrogenation of propane to propene in CO2 assistance (DPH-CO2). In general, the results appear some interest for readers. I recommend publication of the manuscript after major revisions. My suggestion for Authors are reported below.
1. An introduction should contain an information about novelty of this work. Up to know many publications (some reported by authors) are focused on different mesoporous siliceous supports with different pore architecture and porosity. What is new in Authors approach? Pleas explain in introduction.
2. According Authors (introduction) the main goal of the manuscript was investigation of effect of the chromium content on the structure, morphology and catalytic activity. While the catalysts containing 1, 3, 5, 7 and 9 wt.% of Cr was characterized very selectively. For instance, oxidation state of Cr in all fresh catalysts was characterized by UV-Vis DRS, but structure, phase composition, reducibility and porosity only in the case of sample with 9 wt% of Cr and morphology only for the sample with 3 wt. % of Cr. The catalysts with the highest activity (with 5% of Cr) is characterized only by UV-Vis DRS. For correlation with activity H2-TPR and XRD results should be carried out for all Cr loadings, as well. These results give information about presence/absence of inactive crystalline Cr2O3 particles and number of redox Cr sites active in DPH-CO2.
3. How change the catalytic performance with time on stream? Typically, In the DPH-CO2 conversion decrease gradually with time on stream. Which catalysts work the most stable?
4. CO2 is second reactants that is reduced to CO. How change conversion of CO2 vs Cr content? Did Authors observed hydrogen between products?
5. An important problem with CrOx/MCM-41 is stability of mesoporous support in the presence of water produced during this process. In catalytic tests Authors applied vary high temperature (650 and 700 C). How change porosity and chromium dispersion after catalytic process (in air regenerated sample). Please add additional information about catalyst stability under reaction temperature used.
Contradictions and errors:
- Figure 3 is carrier should be support.
- In the text is photographs should be imagines.
- Figure 3D pleas add some color for EDS Cr-mapping.
- TPR are reported between 25-800 C but in text (line 189) is 25-850 C. Pleas correct.
- Table 2 feed proportion need additional information about molar ratio of Propane:CO2:inert (propane conversion changes with this ratio).
Author Response
We are very grateful to the reviewers for their thorough analysis of our manuscript, comments and valuable recommendations. Thanks to the reviewers for the proposed articles, we also included them in the text of the manuscript. We tried to revise the MS according to all the comments on the structure of the manuscript, added the necessary articles and reviews.
We significantly expanded our manuscript by adding various physicochemical data for BET, XRD, SEM-EDS samples, added data on the stability of samples over time and on their regeneration, comparison with literature systems, and a number of other data. Unfortunately, we were not able to investigate all samples with TPR-H2 as we have limited access to this device, but we ask that you re-consider the article with new data, if you think TPR-H2 data are needed, we can add these data within 1-2 weeks.
Reviewer 1
- An introduction should contain an information about novelty of this work. Up to know many publications (some reported by authors) are focused on different mesoporous siliceous supports with different pore architecture and porosity. What is new in Authors approach? Pleas explain in introduction.
The introduction has been improved, the novelty of this study is more clearly formulated.
- According Authors (introduction) the main goal of the manuscript was investigation of effect of the chromium content on the structure, morphology and catalytic activity. While the catalysts containing 1, 3, 5, 7 and 9 wt.% of Cr was characterized very selectively. For instance, oxidation state of Cr in all fresh catalysts was characterized by UV-Vis DRS, but structure, phase composition, reducibility and porosity only in the case of sample with 9 wt% of Cr and morphology only for the sample with 3 wt. % of Cr. The catalysts with the highest activity (with 5% of Cr) is characterized only by UV-Vis DRS. For correlation with activity H2-TPR and XRD results should be carried out for all Cr loadings, as well. These results give information about presence/absence of inactive crystalline Cr2O3 particles and number of redox Cr sites active in DPH-CO2.
The results of the study of texture characteristics (Fig. 1 and Table 1), SEM-EDS (Fig. 3 and Table 2), XRD for all samples (Fig. 4a) have been added to the article. Unfortunately, due to technical reasons, we cannot make H2-TPR for all samples.
- How change the catalytic performance with time on stream? Typically, in the DPH-CO2 conversion decrease gradually with time on stream. Which catalysts work the most stable?
A study of the stability of 5Cr/MCM-41, 9Cr/MCM-41 and 5Cr/SiO2 catalysts has been added to the article (Fig. 11). In addition, the results of catalytic tests after the regeneration of the 5Cr/MCM-41 catalyst are presented.
- CO2 is second reactants that is reduced to CO. How change conversion of CO2 vs Cr content? Did Authors observed hydrogen between products?
The results of CO2 conversion have been added to the article (Figure 7). Hydrogen was not observed.
- An important problem with CrOx/MCM-41 is stability of mesoporous support in the presence of water produced during this process. In catalytic tests Authors applied vary high temperature (650 and 700 C). How change porosity and chromium dispersion after catalytic process (in air regenerated sample). Please add additional information about catalyst stability under reaction temperature used.
A study of the stability of 5Cr/MCM-41, 9Cr/MCM-41 and 5Cr/SiO2 catalysts has been added to the article (Fig. 11).
Contradictions and errors:
-Figure 3 is carrier should be support.
Figure 3 – “carrier” was changed to “support”
-In the text is photographs should be imagines.
In the text, “photographs” were changed to “images”
- Figure 3D pleas add some color for EDS Cr-mapping.
We added some color for EDS Cr-mapping in Figure S1
- TPR are reported between 25-800 C but in text (line 189) is 25-850 C. Pleas correct.
In the text, the range of 25-850 °C was changed to 25-800 °C
- Table 2 feed proportion need additional information about molar ratio of Propane:CO2: inert (propane conversion changes with this ratio).
Table 2. Information on the molar ratio of propane:CO2 was added
Reviewer 2 Report
The manuscript reports the oxidative dehydrogenation of propane by using CO2 as a soft oxidant (CO2-ODP) catalyzed by CrOx/MCM-41. The MCM-41 support was synthesized according to the well known procedure, and characterized by N2 ads/des, TG-DTA, SEM, and XRD. The catalysts with Cr loadings of 1, 3, 5, 7, and 9 were prepared by the incipient impregnation method, and selected samples especially 9Cr/MCM-41 was characterized with different techniques such as XRD and H2-TPR. Even with these characterization results, the chemical & structural information is not as clear as those reported in the references for the same catalysts, e.g., J. Catal., 2004, 224: 404; Appl. Catal. 2008, 349:62, and a recent review work of ACS Catal. 2021, 11, 2182. As for the catalytic results, indeed, many experiments were done. However, without the carbon-balance data of the experiments, the selectivity results are ambiguous especially in the cases of low propane conversions over 1Cr/MCM-41 (Figs. 8 and 9).
Moreover, it is unfair by using Cr/SiO2 as a comparison basis rather than those of the Cr/MCM-41 reported in the references.
Importantly, the key issue of the catalyst for CO2-ODP is the quick deactivation. Unfortunately, there is no any information on the time on stream in the manuscript, let alone the stability results.
One more issue is the analysis of the products in the outlet of the reactor. Is it analyzed on an on-line or off-line GC? What is the analysis error?
As the authors claimed, the purpose of the work is to study the effect of the chromium content on the structure, morphology and catalytic activity of Cr/MCM-41 catalysts. However, with these issues, findings of this work are not consolidated. Moreover, if the reference results of CO2-ODP over CrOx-based (ACS Catal. 2021, 11, 2182) or metal based catalysts (Chem. Eur. J. 2023, 29, e202202173), the catalytic results of this work is also average. Thus, I cannot recommend the acceptance of the work for publication.
Author Response
We are very grateful to the reviewers for their thorough analysis of our manuscript, comments and valuable recommendations. Thanks to the reviewers for the proposed articles, we also included them in the text of the manuscript. We tried to revise the MS according to all the comments on the structure of the manuscript, added the necessary articles and reviews.
We significantly expanded our manuscript by adding various physicochemical data for BET, XRD, SEM-EDS samples, added data on the stability of samples over time and on their regeneration, comparison with literature systems, and a number of other data. Unfortunately, we were not able to investigate all samples with TPR-H2 as we have limited access to this device, but we ask that you re-consider the article with new data, if you think TPR-H2 data are needed, we can add these data within 1-2 weeks.
Reviewer 2
- The MCM-41 support was characterized by N2 ads/des, TG-DTA, SEM, and XRD. The catalysts with Cr loadings of 1, 3, 5, 7 and especially 9Cr/MCM-41 was characterized with different techniques such as XRD and H2-TPR. Even with these characterization results, the chemical & structural information is not as clear as those reported in the references for the same catalysts, e.g., J. Catal., 2004, 224: 404; Appl. Catal. 2008, 349:62, and a recent review work of ACS Catal. 2021, 11, 2182. As for the catalytic results, indeed, many experiments were done. However, without the carbon-balance data of the experiments, the selectivity results are ambiguous especially in the cases of low propane conversions over 1Cr/MCM-41 (Figs. 8 and 9).
Selectivity results for all products for catalysts 1 Cr/MCM-41 and 5 Cr/MCM-41 have been added to the article (Figures 8 and 9)
- Moreover, it is unfair by using Cr/SiO2 as a comparison basis rather than those of the Cr/MCM-41 reported in the references.
Table 3 compares the literature data of catalytic tests on the Cr/MCM-41 catalyst with the results obtained in this work.
- Importantly, the key issue of the catalyst for CO2-ODP is the quick deactivation. Unfortunately, there is no any information on the time on stream in the manuscript, let alone the stability results.
Studies of the lifetime of 5Cr/MCM-41 and 9Cr/MCM-41 catalysts were added to the article (Figure 11). In addition, the results of catalytic tests after regeneration are presented for the 5Cr/MCM-41 catalyst.
- One more issue is the analysis of the products in the outlet of the reactor. Is it analyzed on an on-line or off-line GC? What is the analysis error?
CO2 and C3H8 are supplied to the flow catalytic unit using Bronkhorst EL-FLOW Prestige gas flow regulators, and the gas flow at the reactor outlet is controlled using a SHINAGAWA 2-59853 drum gas meter, which makes it possible to determine the exact space velocity of the gas mixture before and during the catalytic process. The reaction products are analyzed using an interactive gas chromatograph Chromatec-Crystal 5000, which has an absolute calibration for all reaction products, which makes it possible to determine the mole fraction of all components in the gas flow. The relative analysis error is 3-5%
- As the authors claimed, the purpose of the work is to study the effect of the chromium content on the structure, morphology and catalytic activity of Cr/MCM-41 catalysts. However, with these issues, findings of this work are not consolidated. Moreover, if the reference results of CO2-ODP over CrOx-based (ACS Catal. 2021, 11, 2182) or metal based catalysts (Chem. Eur. J. 2023, 29, e202202173), the catalytic results of this work is also average.
Additional results of catalytic tests have been added to the article, such as CO2 conversion (Figure 7), product selectivity for catalysts 1 Cr/MCM-41 (Figure 8) and 5 Cr/MCM-41 (Figure 9), catalyst stability results for 5Cr/MCM-41 and 9Cr/MCM-41 (Figure 11). Additional physico-chemical studies of catalysts have been carried out. The conclusions were finalized in accordance with the obtained results of the catalysis and physico-chemical analysis.
Reviewer 3 Report
There is no doubt that the reaction of propane dehydrogenation is a very interesting topic. And any new and original works in this direction are of great interest. Unfortunately, the presented work is neither new nor original.
The work is devoted to propane dehydrogenation in the presence of CO2 on Cr/MCM-41 catalysts. The same systems have already been used in the same reaction (refs. 40, 41). The novelty of the work is only that the authors use a different propane/CO2 ratio.
The authors write that "aim of the work is to study the effect of the chromium content on the structure, morphology and catalytic activity of Cr/MCM-41 catalysts”. At the same time, studies of structure and morphology are limited to only one sample, and a different sample was used for each method. BJH–BET measurements were carried out only for a single 9Cr/MCM-41 sample, the structure was studied by scanning electron microscopy for a 3Cr/MCM-41 sample, TPR-H2 was carried out only for a 5Cr/MCM-41 sample.
The authors carry out the propane dehydrogenation reaction in the presence of CO2. However, the role of CO2 in this reaction is completely incomprehensible from the results of the work. In order to be clear, it is necessary to carry out the same reaction without CO2. In addition to propane conversion, it is also necessary to determine the conversion and selectivity of CO2.
Describing the data presented in Table 2, the authors write: “It should be noted that although chromium oxide catalytic systems deposited on MCM-41 have lower conversion of propane than the catalytic systems deposited on SiO2, however, they are significantly superior in the selectivity for propylene". In the same time, the table shows that the selectivity for Cr/MCM-41 is 76%, and for Cr/SiO2 it is 100%.
The main task of the work is to study the reaction of propane dehydrogenation in the presence of CO2, while the main conclusions of the work are as follows:
- it is shown that for complete removal of the template it is necessary to use a temperature of 550-600 °C.
- it can be demonstrated that the particles of synthesized MCM-41 have a spherical shape with an average diameter of ~0.8 μm.
- based on SEM-EDS analysis for the 3Cr/MCM-41 catalyst, the Cr content on the surface is shown to be equal to 2.4%.
-the diffraction pattern of the 9Cr/MCM-41 catalyst contains diffraction lines corresponding to the crystalline phase of α-Cr2O3.
There are many inaccuracies and typos in the text, as an example:
- in section 2.6. the caption to figure 7 indicates the 9Cr/MCM-41 sample, while the text describes the 5Cr/MCM-41 sample.
- duplication of the same text: lines 261-263 and 269-271.
Author Response
We are very grateful to the reviewers for their thorough analysis of our manuscript, comments and valuable recommendations. Thanks to the reviewers for the proposed articles, we also included them in the text of the manuscript. We tried to revise the MS according to all the comments on the structure of the manuscript, added the necessary articles and reviews.
We significantly expanded our manuscript by adding various physicochemical data for BET, XRD, SEM-EDS samples, added data on the stability of samples over time and on their regeneration, comparison with literature systems, and a number of other data. Unfortunately, we were not able to investigate all samples with TPR-H2 as we have limited access to this device, but we ask that you re-consider the article with new data, if you think TPR-H2 data are needed, we can add these data within 1-2 weeks.
Reviewer 3
- The same systems have already been used in the same reaction (refs. 40, 41). The novelty of the work is only that the authors use a different propane/CO2
The authors write that "aim of the work is to study the effect of the chromium content on the structure, morphology and catalytic activity of Cr/MCM-41 catalysts”. At the same time, studies of structure and morphology are limited to only one sample, and a different sample was used for each method. BJH–BET measurements were carried out only for a single 9Cr/MCM-41 sample, the structure was studied by scanning electron microscopy for a 3Cr/MCM-41 sample, TPR-H2 was carried out only for a 5Cr/MCM-41 sample.
The results of the study of texture characteristics (Fig. 1 and Table 1), SEM-EDS (Fig. 3 and Table 2), XRD for all samples (Fig. 4a) have been added to the article. Unfortunately, due to technical reasons, we cannot make H2-TPR for all samples.
- The authors carry out the propane dehydrogenation reaction in the presence of CO2. However, the role of CO2 in this reaction is completely incomprehensible from the results of the work. In order to be clear, it is necessary to carry out the same reaction without CO2 In addition to propane conversion, it is also necessary to determine the conversion and selectivity of CO2.
Dehydrogenation of propane is possible without CO2, but with a lower yield of propylene. Our task is to involve CO2 in the chemical process, so all catalytic tests are carried out with the participation of CO2. The results of CO2 conversion are shown in Figure 7a.
- Describing the data presented in Table 2, the authors write: “It should be noted that although chromium oxide catalytic systems deposited on MCM-41 have lower conversion of propane than the catalytic systems deposited on SiO2, however, they are significantly superior in the selectivity for propylene". In the same time, the table shows that the selectivity for Cr/MCM-41 is 76%, and for Cr/SiO2 it is 100%.
It should be noted that although chromium oxide catalytic systems deposited on MCM-41 have a lower conversion of propane than the catalytic systems deposited on SiO2, however, they are significantly superior in the selectivity for propylene, this refers to the description of Figure 11 (in the revised version it is Figure 10), the selectivity on Cr/SiO2 is ~50%.
The main task of the work is to study the reaction of propane dehydrogenation in the presence of CO2, while the main conclusions of the work are as follows:
- it is shown that for complete removal of the template it is necessary to use a temperature of 550-600 °C.
- it can be demonstrated that the particles of synthesized MCM-41 have a spherical shape with an average diameter of ~0.8 μm.
- based on SEM-EDS analysis for the 3Cr/MCM-41 catalyst, the Cr content on the surface is shown to be equal to 2.4%.
-the diffraction pattern of the 9Cr/MCM-41 catalyst contains diffraction lines corresponding to the crystalline phase of α-Cr2O3.
There are many inaccuracies and typos in the text, as an example:
- in section 2.6. the caption to figure 7 indicates the 9Cr/MCM-41 sample, while the text describes the 5Cr/MCM-41 sample.
in the caption to Figure 7, 9Cr/MCM-41 was changed to 5Cr/MCM-41
- duplication of the same text: lines 261-263 and 269-271.
lines 261-263 were removed
Reviewer 4 Report
Comments about the Manuscript ID catalysts-2236602, submitted to Catalysts) and entitled as: Properties of CrOx/MCM-41 and its catalytic activity in the reaction of propane dehydrogenation in the presence of CO2.
Before publication it is very important to clarify the following issues:
1. Please clarify the term SEM-EDM (line 118) or SEM-EDS (line 379)}
2. It is important to complement the information of the SBET, Vmeso and Dpore of all the materials MCM-41, 1Cr/MCM-41, 3Cr/MCM-41, 5Cr/MCM-41, 7Cr/MCM-41, 9Cr/MCM-41.
3. Please specify the type of the Isotherms according to IUPAC.
4. Please include in the discussion the importance of the CrOx clusters in the propane dehydrogenation.
5. If a content of 9% wt. of Co destroyed the mesoporosity of the MCM-41 what is the effect of the lost of the long rang order of the MCM-41 in the dehydrogenation reaction.
6. Please specify the balance between Cr+3 and Cr+6 in the samples.
7. It is important to include in the introduction a Table expressing the main contribution in the literatura in the dehydrogenation of propane expressing the catalyst, temperature of reaction, gas mixture, total flow rate and catalyst loading.
Author Response
Catalysts, Manuscript ID: catalysts-2236602
Title: Properties of CrOx/MCM-41 and its catalytic activity in the reaction of propane dehydrogenation in the presence of CO2
We are very grateful to the reviewers for their thorough analysis of our manuscript, comments and valuable recommendations. Thanks to the reviewers for the proposed articles, we also included them in the text of the manuscript. We tried to revise the MS according to all the comments on the structure of the manuscript, added the necessary articles and reviews.
We significantly expanded our manuscript by adding various physicochemical data for BET, XRD, SEM-EDS samples, added data on the stability of samples over time and on their regeneration, comparison with literature systems, and a number of other data. Unfortunately, we were not able to investigate all samples with TPR-H2 as we have limited access to this device, but we ask that you re-consider the article with new data, if you think TPR-H2 data are needed, we can add these data within 1-2 weeks.
Reviewer 4
- Please clarify the term SEM-EDM (line 118) or SEM-EDS (line 379)}
SEM-EDM stands for scanning electron microscopy with energy dispersion microanalysis. The term SEM-EDS (energy dispersive X-ray spectroscopy) is more often used. Therefore, SEM-EDM changed to SEM-EDS everywhere in the article.
- It is important to complement the information of the SBET, Vmeso and Dpore of all the materials MCM-41, 1Cr/MCM-41, 3Cr/MCM-41, 5Cr/MCM-41, 7Cr/MCM-41, 9Cr/MCM-41.
We added the information on the SBET, Vmeso and Dpore for all samples (table 1, figure 1).
- Please specify the type of the Isotherms according to IUPAC.
All of the silica materials exhibit type IV isotherms according to IUPAC classification. The MCM-41 isotherm shows an H4 type hysteresis loop, indicating the presence of micropores in the material under study.
- Please include in the discussion the importance of the CrOx clusters in the propane dehydrogenation.
We added to the discussion the influence of CrOx on the activity of catalysts in the reaction of propane dehydrogenation in the presence of CO2 (lines 256-258).
- If a content of 9% wt. of Сr destroyed the mesoporosity of the MCM-41 what is the effect of the lost of the long rang order of the MCM-41 in the dehydrogenation reaction.
A consequence of the loss of the MCM-41 long-range order in the 9Cr/MCM-41 sample in the reaction of propane dehydrogenation in the presence of CO2 is a decrease in the activity of the sample, especially during long-term operation, which confirms the study of the stability of this catalyst (Fig. 11).
- Please specify the balance between Cr+3 and Cr+6 in the samples.
Unfortunately, for technical reasons, we cannot produce H2-TPR for all samples, therefore we indicate the balance of Cr3+ and Cr6+ species in the samples.
- It is important to include in the introduction a Table expressing the main contribution in the literatura in the dehydrogenation of propane expressing the catalyst, temperature of reaction, gas mixture, total flow rate and catalyst loading.
A comparison table with literature data and our results was added to results and discussions (Table 3)
Round 2
Reviewer 1 Report
Authors improved the manuscript according to the suggestions. Thus, I recommend this manuscript for publication.
Reviewer 2 Report
Indeed, the quality of the manuscript is improved after revision and adding some new results. Hower, in comparison with those of the published works of the subject, the manuscript is still inferior. Sorry that I cannot give a positive recommendation for the publication of the work.
Reviewer 3 Report
The authors took into account the main comments of the reviewers. The revised manuscript may be accepted for publication in the journal.
Reviewer 4 Report
The authors revised properly the manuscript.